# Parenting Stress, Parent–Child Literacy Activities, and Pre-Schoolers' Reading Interest: The Moderation Role of Child Number in Chinese Families

**Jia Yang [1], Wanlin Xie [2], Xunyi Lin [2,*] and Hui Li [3]**

[1]   Department of Early Childhood Education, Fujian Preschool Education College, Fuzhou 350007, China
[2]   College of Education, Fujian Normal University, Fuzhou 350007, China
[3]   Shanghai Institute of Early Childhood Education, Shanghai Normal University, Shanghai 200234, China
*   Correspondence: xunyilin@fjnu.edu.cn

**Abstract:** China replaced the Two-Child Policy with the Three-Child Policy in 2021 to raise birth rates, but the potential effects of the increased number of children on family life and child development have not been empirically explored. This study examines the moderating role of child number in the relationships between parenting stress, parent–child literacy activities, and young children's reading interest in the new Three-Child Policy context in China. A sample of 895 Chinese families was randomly recruited from a coastal city in southeastern China: one-child families ($N_{one-child}$ = 359, $M_{age}$ = 5.0, $SD$ = 0.9), two-child families ($N_{two-child}$ = 469, $M_{age}$ = 5.1, $SD$ = 0.9), and three-child and above families ($N_{three-child\ and\ above}$ = 67, $M_{age}$ = 5.2, $SD$ = 1.0). The participants completed the Parenting Stress Index-Short Form, the Parent-Child Literacy Activities Scale, and the Children's Reading Interest Questionnaire. The results showed that (1) parents with more children had higher levels of parenting stress; (2) no significant differences existed in children's reading interest between families with different numbers of children; (3) parent–child literacy activities mediated the relationship between parenting stress and children's reading interest; (4) child number moderated the mediating effect of parent–child literacy activities in the relationship between parenting stress and children's reading interest, even after adjusting for child age, gender, and family socioeconomic status (SES). Overall, this study demonstrated how the increased number of children would interact with the family system concerning early childhood literacy development.

**Keywords:** parenting stress; home literacy activities; children's reading interest; child number

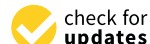



## 1. Introduction

China launched the Two-Child policy in 2016 [1] and the Three-Child Policy in 2021 to cope with the challenge of an aging population caused by the lowest low fertility since the turn of this millennium. China's aging population has threatened the country's sustainable development, leading China's central government to radically change its fertility policy and try all measures to encourage young couples to give birth to more children [2]. However, young parents with one or two children are very reluctant to have a third. The latest national census released by the National Bureau of Statistics (2021) has revealed that there were 45.8% one-child families, 43.1% two-child families, and 11.1% three-child and above families in 2020 in China [3]. Regarding maternal education, 66.6% of mothers had a high school or higher education in one-child families, 46.4% of mothers had a primary or junior high school education in two-child families, and 60.0% of mothers had a junior high school education in three-child and above families [3]. It is widely believed that the number of children plays a decisive role in parenting, family functioning, and child development [4–7]. For example, according to the Quantity–Quality theory [8], having more children in a family would cause increased parenting stress and lower the quality per child as resources are diluted [4]. In addition, more children would bring

more challenges to parents and make sibling interactions more complex [9]. Child policy transformation as a macrosystem-level change would impact the innermost microsystem, such as parenting and family relationships. However, no existing studies have thoroughly explored the potential impact of the increased number of children on parenting stress and home learning environment in the new Three-Child Policy context. Children learn and develop only through activities with others or interactive patterns. To fill this gap, the present study examined the possible mediating/moderating effects of child number and parent–child activities on the relationship between parenting stress and child reading interest in Chinese families.

### 1.1. Parenting Stress, Parent–Child Literacy Activities, and Children's Reading Interest

Numerous studies have confirmed a significant association between parenting quality and socio-cognitive development in early childhood. Lareau (2003) argued that social class inequality would cause differences in parenting quality across families [10]. Specifically, parents with a sense of powerlessness and frustration in parenting may contribute to low-level parenting involvement in education, which is harmful to early childhood development. Parenting stress refers to negative feelings toward the parent/self and a sense of pressure directly related to raising children [11]. This stress can often disrupt parents' peaceful lives, hinder their construction of a good home environment, and likely result in parents' poor mental health and children's negative development [12,13]. Drawing on the theoretical perspectives of the Family Stress Model (FSM), family hardships may increase parents' emotional distress and hurt family life and parenting practices [14], which in turn detrimentally influences children's cognitive development [15]. A six-year longitudinal study reported that parenting stress would predict young children's internalising problems [16]. Children with highly depressed parents have more difficulties interacting with their parents and other children, which could cause them to have more behavioural problems and poorer cognitive and language development outcomes [17–19]. Thus, the present study examined whether parenting stress directly or indirectly affected child development in reading and literacy.

In light of the Family Investment Model (FIM), parents' capacity to invest sufficient and high-quality time (e.g., spending time reading or discussing with children) and beneficial resources (e.g., social contacts or reading resources) into child-rearing is closely linked to high parenting quality with positive consequences on children's learning motivation and cognitive achievement [14,20]. Parent–child literacy activities refer to parents reading to their children at home and discussing the reading with them [21]. These home-based activities and interactions provide a critical environment for developing early child language skills [22,23]. Parent–child literacy activities that foster high-quality verbal interactions between parents and their children offer lexical richness [24] and are conducive to the affective quality of the home literacy environment [25], both of which influence children's literacy development [26]. Children become more engaged when they and their parents share positive interactions while reading books, enhancing their early literacy skills [27]. However, when parents feel high levels of stress when raising their children, they may decrease their involvement in their children's literacy activities, as parenting stress is negatively associated with both the quantity and quality of parents' interactions with their children [19,26,28].

Children's reading interest is a positive attitude towards reading, influencing their willingness to engage in it [29]. Baker et al., (1997) proposed that children's reading interests could include affective, cognitive, and behavioural components and be a significant aspect of children's literacy development [30]. Previous studies indicate that mothers' perceived parenting stress negatively affects pre-schoolers' language development and literacy skills [18,19,31]. However, there is a dearth of research on the relationship between parenting stress and young children's attitudes, such as children's interest in reading. Moreover, there is a need to examine the possible mechanisms underlying the link between parenting stress and children's attitudes toward reading to shed light on improving parenting prac-

tices. Therefore, this study examined the potential mediation role of parent–child activities on literacy in the relationship between parenting stress and children's reading interest.

## 1.2. The Number of Children in Family

China launched its Three-Child Policy in 2021 to boost birth rates and solve the problems of an aging society. This new population policy is highly significant and has inspired many scholars to intensively study its broader impacts on the family structure and child development [32]. There have been considerable debates on the differences in psychological status and behaviours of parents and children among families with different numbers of children, and the empirical evidence is mixed. On the one hand, Qian et al. (2021) reported that mothers with two children had higher parenting stress than mothers with only one child [33]. Regarding parenting language-learning children, the number of children in a family (e.g., children with siblings) was significantly negatively associated with the number of parent–child interactions at home [34]. On the other hand, research has demonstrated that sibling relationships contribute to children's cognitive, linguistic, and socio-emotional development [35,36]. The sibling subsystem within a family has a unique and powerful influence on how the family functions, promoting or detracting from parents' efforts to socialise with their children [36]. Therefore, this study empirically examined the potential impact of the increased number of children in a family on Chinese parents and children in the new Three-Child Policy context by modelling the mediating/moderating roles of child number on the relations between parenting stress, parent–child literacy activities, and children's reading interest.

## 1.3. The Present Study

According to ecological systems theory [37,38], how a child develops is affected by an integrated system with multiple levels of the surrounding environment, from the most immediate microsystem (e.g., family relationships) to the broadest macrosystem (e.g., cultures, laws, and policies). In this study, we operationalised implementing the Three-Child Policy as a macrosystem-level change that would impact the innermost microsystem. Therefore, we examined macro-contextual changes that would interact with family systems concerning early child development. Family Systems Theory (FST) emphasises family functioning as a major force contributing to adaptive or maladaptive child development [39]. FST also views different levels of social-ecological influences on children's development through the activities in which children engage. Children can learn and develop only through activities with others or children's interaction patterns [40]. From this perspective, parenting stress might directly affect parenting behaviours and children's development. The present study examined parent–child literacy activities occurring within the family as the mediating factor in the relationship between parenting stress and children's literacy development. Based on a review of Bronfenbrenner's theory [37,38] and the FST perspective [39,40], we proposed a conceptual model, as displayed in Figure 1, for predicting early childhood literacy development in the context of the Three-Child Policy.

Accordingly, the following research questions guide this study:

(1) Are there any differences in parenting stress, children's reading interest, and parent–child literacy activities among families with different numbers of children?
(2) Does parenting stress negatively predict children's reading interest?
(3) Do parent–child literacy activities mediate the relationships between parenting stress and children's reading interest?
(4) Does child number moderate the relationship between parenting stress and children's reading interest, and the mediating relations between parenting stress and children's reading interest via parent–child literacy activities?

The present study hypothesizes parenting stress as an extrinsic variable that may be directly or indirectly linked to children's literacy development (i.e., reduced children's reading interest), as existing literature has demonstrated the association between high-level stress and low-level literacy [18,19,31] and the mediating effects of parent–child interactions

on the relationship between parenting stress and child's reading development [15,26,28]. Furthermore, given that more children in a family may be linked to higher levels of parenting stress [4,33,41] and reduce parents' capacity to invest high-quality time and resources per child [14,20,42,43], it is reasonable to hypothesize that the number of children in a family may moderate the association between parenting stress and child development and the mediating effect of parent–child interaction. Taken together, the following hypotheses were made and tested:

**Hypothesis 1 (H1).** *There would be differences in parenting stress among families with different numbers of children.*

**Hypothesis 2 (H2).** *There would be no differences in children's reading interests and parent–child literacy activities among families with different numbers of children.*

**Hypothesis 3 (H3).** *Parenting stress would negatively predict children's reading interest, even controlling for child age, child gender, parent gender, and SES.*

**Hypothesis 4 (H4).** *Parent–child literacy activities would mediate the relationship between parenting stress and children's reading interest, even controlling for child age, child gender, parent gender, and SES.*

**Hypothesis 5 (H5).** *Child number would moderate the relation between parenting stress and children's reading interest, even controlling for child age, child gender, parent gender, and SES.*

**Hypothesis 6 (H6).** *Child number would moderate the mediating effect of parent–child literacy activities on the relationship between parenting stress and children's reading interest, even controlling for child age, child gender, parent gender, and SES.*

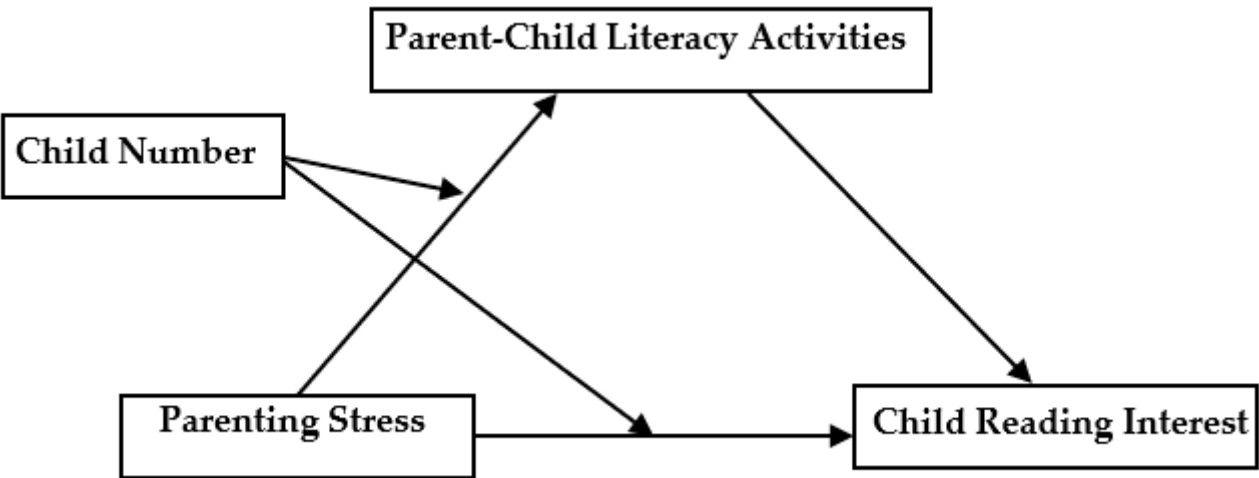

**Figure 1.** The proposed conceptual model for child number in family, parenting stress, parent–child literacy activities, and children's reading interest.

## 2. Materials and Methods

### 2.1. Participant and Procedures

This study recruited the participating children and their parents from three public kindergartens in a large city in southeastern China. According to the local Kindergarten Rating Assessment Program, two kindergartens in this study were rated as "Municipal Demonstration Kindergarten", and one kindergarten was "not rated". In addition, there are two kindergartens in an urban area and one in a rural area. A total of 895 children and their parents participated in this study. The sample consisted of 359 (40.1%) children from one-child families, 469 (52.4%) children from two-child families, and 67 (7.5%) children

from three-child and above families. These children and their parents were recruited via a convenience sampling approach. There were 186 boys and 173 girls in one-child families, with an average age of 5.0 years (*SD* = 0.9); 264 boys and 205 girls in two-child families, with an average age of 5.1 years (*SD* = 0.9); and 38 boys and 29 girls in three-child and above families, with an average age of 5.2 years (*SD* = 1.0). Regarding parents' education, 22.8% of fathers and 21.8% of mothers had a high school or lower education, and most parents (49.9% of fathers and 50.0% of mothers) had a bachelor's degree or higher. Most fathers (53.3%) and mothers (36.2%) reported their vocation as professionals or officers. Annual family income (RMB) was measured categorically and distributed as follows: below 80,000 (3.1%), 80,000–150,000 (16.4%), 150,000–300,000 (41.3%), 300,000–1,000,000 (33.5%), and over 1,000,000 (5.7%).

The participants were informed of the purpose of the study, advised that their participation was voluntary, and told that they could withdraw any time they wished. The participants provided their written informed consent to participate in this study. Next, an online survey link was shared with all participating parents, who completed the questionnaires independently via the Internet. The online survey lasted for approximately 15 min. No personal identifiers have been kept or recorded in this study, and no privacy risks could be caused. Therefore, ethical review and approval were not required per the local institutional requirements. After completing the survey, the first and second authors downloaded, entered, and analysed all the data.

### 2.2. Measures

### 2.2.1. Demographic Questionnaire

The purpose of the demographic survey was to acquire the participants' background information, including the child's age and gender, the number of children in each family, and parents' information, including parental education, occupation, and household income. We adopted three widely-used markers of socioeconomic status (SES): household income (estimated gross annual income), maternal and paternal education, and maternal and paternal occupation. Annual household income was measured with a five-point scale (1 = less than RMB 80,000 to 5 = RMB 1,000,000 and above), parental education was measured with a four-point scale (1 = high school and below to 4 = master's degree and above), and parental occupation was measured with a five-point scale (1 = stay-at-home or unemployed to 5 = high-level professional and administrator). Family income, maternal and paternal education, and maternal and paternal occupations were standardised (Z-score), and then SES was identified as an observed construct using five standard scores [44].

### 2.2.2. Parenting Stress

Parenting stress was assessed using the Parenting Stress Index-Short Form (PSI-SF) [13]. It was verified that PSI-SF was applicable to parents of preschool children in the Chinese sample [45]. The PSI-SF contains 36 items divided into three 12-item subscales: parental distress (e.g., "I sacrifice a lot of my life to meet the needs of this child."), parent–child dysfunctional interaction (e.g., "My child does things deliberately that make me very angry."), and difficult child (e.g., "it is more difficult than I expected for my child to develop a regular routine because he/she does not sleep or eat regularly"). Parents responded on a five-point Likert scale ranging from 1 (strongly disagree) to 5 (strongly agree). Cronbach's $\alpha$ for the PSI-SF was 0.95 in the current study.

### 2.2.3. Parent–Child Literacy Activities

The parent–child literacy activities scale comes from the learning-to-read survey used in the Progress in International Reading Literacy Study (PIRLS) [46]. In the previous study, these items were used to examine the home literacy environment of Chinese children [47–49] and were tested to apply to the Chinese context. The parent–child literacy activities scale consisted of four items using a three-point scale ranging from 1 (never or almost never) to 3 (frequently). The items contained the frequency of (a) sharing book

reading, (b) telling stories, (c) teaching a child to write characters, and (d) teaching a child to read words on signs and labels. In the present study, Cronbach's $\alpha$ for the parent–child literacy activities scale was 0.75.

### 2.2.4. Children's Reading Interest

Parents completed the children's reading interest questionnaire to report their children's reading interests. The children's reading interest questionnaire was adapted from the parental questionnaire used in the Program for International Student Assessment (PISA) [50]. Considering the reading level of the participants in the present study, an item about reading to collect information was deleted. The children's reading interest questionnaire consisted of 10 items (e.g., "He/she likes to go to bookstores or libraries") using a four-point Likert scale ranging from 1 (strongly disagree) to 4 (strongly agree). Cronbach's $\alpha$ for the children's reading interest scale was 0.83 in this study.

### 2.3. Data Analyses

There were no missing responses in the dataset. IBM SPSS-23 was used to save and analyse all the data. First, Pearson correlation analyses were conducted to explore the study variables' associations. Second, a set of ANOVAs was conducted to examine the number of children in the family's effect on parenting stress, parent–child literacy activities, and children's reading interest. Third, based on the correlations of the variables, a bootstrapping analysis with 5000 resamples was conducted to test the moderated mediating model using SPSS macro-PROCESS 2.1 [51].

## 3. Results

Harman's single-factor test was first carried out on the Parenting Stress Index-Short Form, parent–child literacy activities scale, and children's reading interest questionnaire. The results showed that the eigenvalues of the three factors were greater than 1. The interpretation rate of the first common factor is 23.97%, which is less than 40% of the critical standard, indicating that there is no serious Common Method Bias (CMB) in this study.

### 3.1. Descriptive Statistics

Table 1 presents the descriptive statistics and correlation analysis, which revealed that SES was negatively related to the number of children in a family ($r = -0.208$, $p < 0.01$) and parenting stress ($r = -0.134$, $p < 0.01$). Results also indicated that SES was positively related to parent–child literacy activities ($r = 0.131$, $p < 0.01$) and children's reading interest ($r = 0.079$, $p < 0.05$). The mean score of parenting stress ($M = 1.894$, $SD = 0.586$) was positively correlated with the number of children ($r = 0.071$, $p < 0.05$), while parenting stress was negatively related to parent–child literacy activities ($r = -0.171$, $p < 0.01$) and children's reading interest ($r = -0.320$, $p < 0.01$). Children's reading interest was significantly related to parent–child literacy activities ($r = 0.379$, $p < 0.01$).

**Table 1.** Means, standard deviations, and correlations of all variables in this study.

| | *M* | *SD* | 1 | 2 | 3 | 4 | 5 | 6 | 7 | 8 |
|---|---|---|---|---|---|---|---|---|---|---|
| 1. Child gender | – | – | – | | | | | | | |
| 2. Child age | – | – | 0.026 | – | | | | | | |
| 3. Parent gender | – | – | 0.042 | 0.000 | – | | | | | |
| 4. SES | – | – | 0.018 | −0.067 * | −0.029 | – | | | | |
| 5. Number of children | – | – | −0.041 | 0.055 | −0.015 | −0.208 ** | – | | | |
| 6. Parenting stress | 1.894 | 0.586 | 0.031 | −0.028 | 0.022 | −0.134 ** | 0.071 * | – | | |
| 7. Parent–child literacy activities | 2.341 | 0.424 | −0.025 | 0.060 | 0.011 | 0.131 ** | −0.056 | −0.171 ** | – | |
| 8. Children's reading interest | 3.063 | 0.449 | 0.034 | 0.011 | 0.039 | 0.079 * | −0.023 | −0.320 ** | 0.379 ** | – |

\* $p < 0.05$; \*\* $p < 0.01$.

### 3.2. Singleton's Effect on Parenting Stress, Parent–Child Literacy Activities, and Children's Reading Interest

ANOVAs were conducted to answer research questions 1 and 2, with the number of children in the family as the independent variable and parenting stress, parent–child literacy activities, and children's reading interest as the dependent variables. Results showed that parenting stress ($p < 0.05$) differed as a function of the number of children in families (see Table 2). Families with more children scored higher for parenting stress than families with fewer children. Results also revealed no significant difference in parent–child literacy activities and children's reading interest among families with different numbers of children ($p > 0.05$). Thus, both H1 and H2 were supported.

**Table 2.** Results from one-way ANOVAs: parenting stress, parent–child literacy activities, and children's reading interest as a function of the number of children in families.

|  | Number of Children | | | |
|---|---|---|---|---|
|  | One Child | Two Children | Three Children and above | *F* Value |
| Parenting stress | 1.83(0.55) | 1.94(0.61) | 1.91(0.61) | 3.223 * |
| Parent–child literacy activities | 2.37(0.41) | 2.32(0.43) | 2.32(0.48) | 1.657 |
| Children's reading interest | 3.08(0.47) | 3.05(0.44) | 3.06(0.44) | 0.374 |

* $p < 0.05$.

### 3.3. The Mediating Effect of Parent–Child Literacy Activities

To answer research questions 3 and 4, regression analyses were employed to explore how parenting stress and parent–child literacy activities were associated with children's reading interests. As shown in Table 3, parenting stress was significantly predictive of children's reading interest, explaining 10.8% of the variance ($R^2 = 0.108$) in the frequency of reading interest ($F = 21.461$, $p < 0.001$), along with the control variables (child age, child gender, parent gender, and SES). Parenting stress was negatively associated with parent–child literacy activities ($F = 8.583$, $p < 0.001$), explaining 4.6% of the variance ($R^2 = 0.046$). Model 3 showed that parenting stress and parent–child literacy activities collectively accounted for 21.5% of the variance ($R^2 = 0.215$) in the frequency of children's reading interest ($F = 40.592$, $p < 0.001$). These results indicated that parent–child literacy activities significantly mediate the relationship between parenting stress and children's reading interest. Thus, both H3 and H4 were supported.

**Table 3.** Regression results for simple mediation.

| Dependent Variable | Independent Variable | *B* | *SEB* | *t* | $R^2$ | $\Delta R^2$ |
|---|---|---|---|---|---|---|
| Model 1 |  |  |  |  |  |  |
| Children's reading interest |  |  |  |  | 0.108 | 0.103 |
|  | Constant | 3.380 | 0.120 | 28.182 *** |  |  |
|  | Child gender | 0.037 | 0.029 | 1.286 |  |  |
|  | Child age | 0.002 | 0.016 | 0.103 |  |  |
|  | Parent gender | 0.042 | 0.029 | 1.441 |  |  |
|  | SES | 0.003 | 0.003 | 1.172 |  |  |
|  | Parenting stress | −0.243 | 0.025 | −9.908 *** |  |  |
| Model 2 |  |  |  |  |  |  |
| Parent–child literacy activities |  |  |  |  | 0.046 | 0.041 |
|  | Constant | 2.385 | 0.117 | 20.336 *** |  |  |
|  | Child gender | −0.022 | 0.028 | −0.770 |  |  |
|  | Child age | 0.030 | 0.016 | 10.957 |  |  |
|  | Parent gender | 0.017 | 0.029 | 0.584 |  |  |
|  | SES | 0.010 | 0.003 | 3.503 ** |  |  |
|  | Parenting stress | −0.111 | 0.024 | −4.617 *** |  |  |

**Table 3.** *Cont.*

| Dependent Variable | Independent Variable | B | SEB | t | R² | ΔR² |
|---|---|---|---|---|---|---|
| Model 3 | | | | | | |
| Children's reading interest | | | | | 0.215 | 0.210 |
| | Constant | 2.534 | 0.136 | 18.597 *** | | |
| | Child gender | 0.044 | 0.027 | 1.655 | | |
| | Child age | −0.009 | 0.015 | −0.613 | | |
| | Parent gender | 0.036 | 0.028 | 1.320 | | |
| | SES | 0.000 | 0.003 | −0.047 | | |
| | Parenting stress | −0.204 | 0.023 | −8.746 *** | | |
| | Parent–child literacy activities | 0.355 | 0.032 | 11.031 *** | | |

In each model, child age, child gender, parent gender, and SES were control variables entered in the first step. ** $p < 0.01$; *** $p < 0.001$.

### 3.4. The Moderated Mediating Effects

To answer research questions 5 and 6, we conducted the moderated mediating analysis using the PROCESS macro (model 8) [52,53]. In addition, we added child age, child gender, parent gender, and SES as the control variables. As Table 4 shows, the interaction of parenting stress and the number of children in the family significantly affected children's reading interest ($B = 0.088$, $p < 0.05$) and parent–child literacy activities ($B = −0.115$, $p < 0.05$). Bootstrap CI and index of moderated mediation corroborated both findings.

**Table 4.** Regression results for moderated mediation.

| Predictor | B | SE | t | p |
|---|---|---|---|---|
| Parent–child literacy activities | | | | |
| Constant | 2.240 | 0.132 | 17.012 | 0.000 |
| Child gender | −0.026 | 0.028 | −0.914 | 0.361 |
| Child age | −0.033 | 0.016 | 2.101 | 0.036 |
| Parent gender | 0.013 | 0.029 | 0.459 | 0.647 |
| SES | 0.009 | 0.003 | 3.163 | 0.002 |
| Parenting stress | −0.022 | 0.040 | −0.534 | 0.594 |
| Number of children | 0.223 | 0.097 | 2.289 | 0.022 |
| Parenting stress × Number of children | −0.135 | 0.050 | −2.706 | 0.007 |
| Children's reading interest | | | | |
| Constant | 2.640 | 0.146 | 18.116 | 0.000 |
| Child gender | 0.048 | 0.027 | 1.775 | 0.076 |
| Child age | −0.011 | 0.015 | −0.741 | 0.459 |
| Parent gender | 0.039 | 0.028 | 1.419 | 0.156 |
| SES | 0.000 | 0.003 | 0.154 | 0.878 |
| Parenting stress | −0.277 | 0.039 | −7.157 | 0.000 |
| Parent–child literacy activities | 0.363 | 0.032 | 11.236 | 0.000 |
| Number of children | −0.193 | 0.094 | −2.054 | 0.040 |
| Parenting stress × Number of children | 0.113 | 0.048 | 2.343 | 0.019 |
| Index of moderated mediation | | | | |
| | Index | Boot SE | Boot LLCI | Boot ULCI |
| Number of children | −0.049 | 0.020 | −0.090 | −0.012 |

In each model, child age, child gender, parent gender, and SES were entered as the control variables. N = 895 Bootstrap sample size = 5000. LLCI = lower bound in 95% confidence interval; ULCI = upper bound in 95% confidence interval.

We explained the significant interaction by plotting a simple slope. As shown in Figures 2 and 3, the slope of the relationship between parenting stress and children's reading interest was relatively strong for only children ($t = −7.157$, $p < 0.001$) and relatively weak for sibling children ($t = −5.663$, $p < 0.001$). On the contrary, the slope of the relationship between parenting stress and parent–child literacy activities was relatively strong for sibling

children ($t = -5.275$, $p < 0.001$) and relatively weak for only children ($t = -0.534$, $p > 0.05$). Thus, both H5 and H6 were supported.

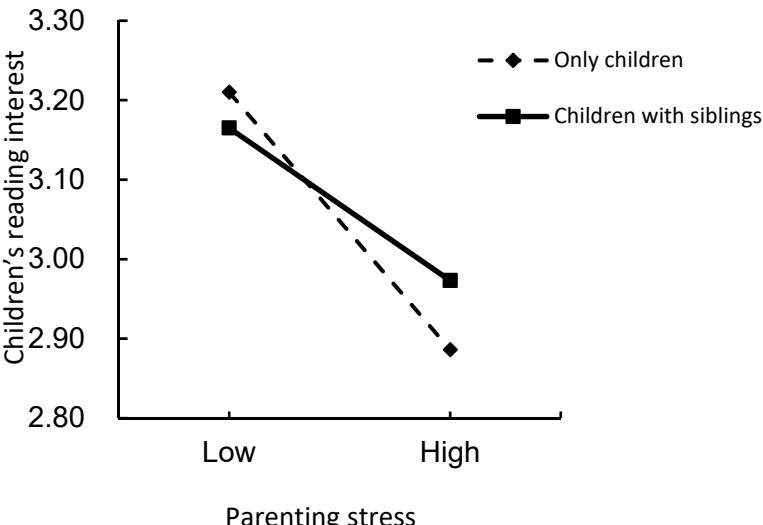

**Figure 2.** Child number moderates the relationship between parenting stress and children's reading interest.

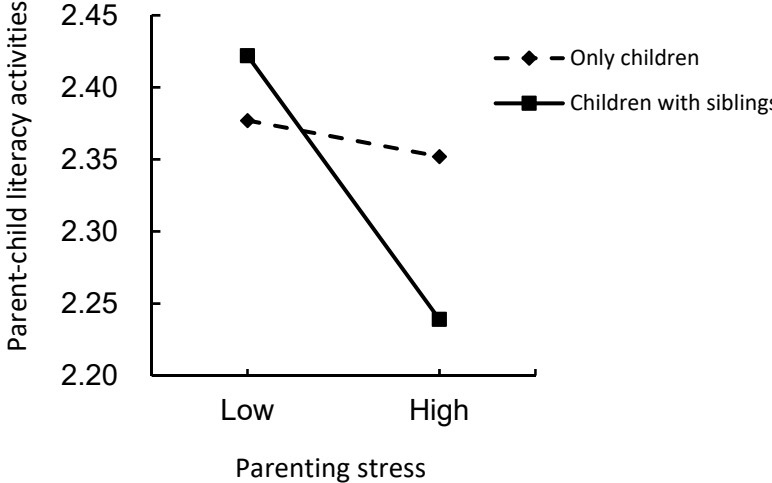

**Figure 3.** Child number moderates the relationship between parenting stress and parent–child literacy activities.

## 4. Discussion

This study revealed the mediating effect of parent–child literacy activities and the moderating effect of child number on the association between parenting stress and children's reading interest. Moreover, the results demonstrated that the mediating effect of parent–child literacy activities varied at different values of the number of children in the family as the moderator. These findings and their implications are discussed in the Three-Child Policy context.

### 4.1. Parenting Stress with Different Numbers of Children

Significant differences were found in this study in parenting stress among families with different numbers of children, supporting Hypothesis 1. Specifically, the parents with more children had higher levels of parenting stress. The finding is consistent with previous studies showing that the number of children in the family directly impacted

both mothers' and fathers' levels of distress [54]. Hong and Liu (2019) reported that, in China, parenting stress is significantly higher in a two-child family than in a single-child family [55]. Bossard's (1945) law of family interaction suggests that family relations will be more complex and diverse in multi-child families than in single-child families [56]. Parenting more children requires spending more time and energy adapting to different child personalities and facing new, stressful challenges and complex situations, such as parent–child relationships [57,58] and sibling relationships [59].

No differences were found in this study in children's reading interest and parent–child literacy activities among families with different numbers of children, supporting Hypothesis 2. However, there is considerable debate concerning the impact of the number of children in a family on children's reading interests and parent–child literacy activities. This study's results were in accordance with Bracken and Fischel's (2008) finding that the number of children in the family was not significantly related to child reading interest and parent–child reading interaction [60]. However, Hart and Risley (1992) reported that parents' engagement in children's literacy activities would decrease as the number of children increased in the family [34]. Therefore, based on the previous debates [34,60] and the results of this study, we believe that children's reading interests and parent–child literacy activities in families with different numbers of children might be affected by multiple factors.

### 4.2. Direct and Indirect Effects of Parenting Stress on Children's Reading Interest

This study showed that parenting stress could negatively predict reading interest in young children after controlling for child age, gender, parent gender, and SES, providing empirical evidence supporting Hypothesis 3. This corroborates previous studies [19,31,61], emphasising associations of parenting stress with economic difficulty, mental health, and psychological problems, which could directly impact parents' responsiveness to their children and the likelihood of a supportive and conducive home environment provided for children's literacy development.

Furthermore, the mediation analyses revealed that parent–child literacy activities mediated the relationship between parenting stress and children's reading interest after controlling for child age, gender, parent gender, and SES, as predicted in Hypothesis 4. This finding is consistent with previous research that associated children's literacy interest with parents' direct involvement in and encouragement of literacy-related activities [62,63] and mothers' perceived parenting stress levels [31]. This result also aligns with Weigel et al.'s (2010) finding that higher parenting stress was associated with lower parent–child literacy activities, and parents' efforts to directly engage children in literacy activities were associated with children's increased interest in reading [19].

Thus, the present study's findings on the significant association between parenting stress, parent–child literacy activities, and children's reading interest correspond to the existing studies in the literature. In addition, these results suggested that a parenting system in the home environment was the primary and powerful influence, directly and indirectly, related to early childhood literacy development.

### 4.3. Child Number as the Moderator

This study has confirmed the moderated mediation model proposed after controlling for child age, gender, parent gender, and SES, providing partial support for H5 and H6. In particular, the effect of parenting stress on children's reading interest and the mediating effect of parent–child literacy activities were moderated by child number in the family, even after controlling for child age, gender, and SES. These findings coincide with previous studies indicating the critical role of child number in a family on the relationship between parenting stress, parent–child interactions, and children's literacy development [33,34], the assertions of FST, and Bronfenbrenner's (1986) ecological systems approach [64]. These findings suggest that negative parenting system factors, such as parenting stress, would contribute to different levels of parent–child interaction (e.g., parent–child literacy activities)

and children's literacy development (e.g., children's reading interests). Having more children could increase the negative effect of parenting stress on parent–child literacy activities and diminish parenting stress's negative effects on children's reading interests.

Specifically, the moderating effect of the number of children in the family on the relationship between parenting stress and parent–child literacy activities and parenting stress and children's reading interest could be explained. On the one hand, adding family members requires reorganisation of the family system [43]. In addition, parenting more children requires more adjustments, and parents generally lack experience raising multiple children [55]. Thus, parents raising more children would experience higher parenting stress. On the other hand, the transactional perspective emphasises the interdependencies among social systems within the family (i.e., parent–child and the sibling subsystem) and family members' factors (i.e., parenting stress and children's reading interest) and how both affect and are affected by events occurring in other subsystems [64]. In particular, sibling relationships serve as a fundamental part of the family functioning and dynamics, providing an important context for learning and conversing. The existing literature has explored the significant role young siblings play in each other's literacy development, arguing that there is a unique reciprocity (synergy) in young children's learning, with siblings acting as facilitators to stimulate and promote each other's literacy development [65]. Segal et al. (2017) demonstrated that siblings had as much enthusiasm as each other's teachers for learning literacy concepts and skills during naturalistic interactions in the home, indicating that siblings would enrich the home literacy environment and further increase children's literacy interest [66]. Thus, positive interactions between siblings could act as a buffer, alleviating parenting stress's negative impact on young children's reading interests.

### 4.4. Limitations, Future Directions, and Implications

This study has some limitations that need to be improved in future studies. First, the participants in this study were all from the same city. This can affect the extent to which the findings are generalisable. Future research should expand the source of participants, select families from other cities or the whole country to participate in the study, and improve the applicability of the research results. In addition, the questionnaire for this study was distributed to parents via the Internet and completed independently; thus, there was no way to avoid socially desirable bias. Finally, future studies should include the investigator's assessment of the participant's background information to reduce errors in the research results.

Despite its limitations, this study has theoretical implications. Based on family system theories and Bronfenbrenner's (1986) ecological systems approach, we proposed and tested a model that simultaneously examined parent–child literacy activities as a mediating mechanism and child number as a moderating factor in the associations between parenting stress, parent–child literacy activities, and children's reading interest in the context of the Three-Child Policy, and found parent–child reading activities could enhance children's interest in reading.

This study also has practical implications. It has shown that parenting stress adversely affects parent–child literacy activities and children's reading interests; lessened parenting stress is more conducive to parent–child literacy interaction and children's literacy development. Furthermore, in identifying the moderating role of the number of children in the family on the mediating process of parenting on literacy development, we suggest that, under the Three-Child Policy, local family education guidance centers and schools should provide parents with appropriate methods to reduce parenting stress and professional guidance to cope with parenting difficulties. Above all, parents should attempt to treat child-rearing difficulties correctly and continually improve their upbringing skills to reduce parenting stress, increase their active participation in their children's literacy activities, and promote effective interaction between siblings to enhance children's interest in reading. Only when these problems are solved would Chinese parents consider having more children, contributing to the sustainable development of an aging country.

**Author Contributions:** J.Y. co-designed the research, collected the data, and drafted the manuscript; W.X. conducted the statistical analyses and drafted the manuscript; X.L. determined the research questions and focus, co-designed the research, and drafted the manuscript. H.L. provided important ideas and substantial feedback to the study and edited the manuscript. All authors have read and agreed to the published version of the manuscript.

**Funding:** This research was funded by the National Education Sciences Planning Fund of China, grant number BDA210076.

**Institutional Review Board Statement:** Ethical review and approval were not required for the study on human participants in accordance with the local legislation and institutional requirements. The patients/participants provided their written informed consent to participate in this study. Participants were also advised that their participation was purely voluntary, and they could withdraw anytime if they wished to do so. Since no personal identifiers have been kept or recorded in the database, no privacy, the conclusions of this article will be made available by the authors without undue reservation.

**Informed Consent Statement:** Informed consent was obtained from all subjects involved in the study.

**Data Availability Statement:** The raw data supporting the conclusions of this article will be made available by the authors without undue reservation.

**Conflicts of Interest:** The authors declare no conflict of interest.

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
