# Peer review of "Parenting Stress, Parent–Child Literacy Activities, and Pre-Schoolers’ Reading Interest: The Moderation Role of Child Number in Chinese Families"

_sustainability, doi:10.3390/su142315783_

Round 1
Reviewer 1 Report
I compliment the authors for an interesting article. The study examines the relationship between parenting stress, parent child literacy activities and the reading interest of children. It particularly focuses on the moderating role of the number of children in the Chinese context. Below I provide some suggestions that go in the direction of further improvement before the publication.
My first suggestion would be to move from the focus on three-child policy. I found it misleading to look at the article as an evaluation of this new measure of Chinese Government. In other words, the authors indicate that they study the effect of three child policy (“Therefore, this study 96 empirically examined the Three-Child Policy’s effect on Chinese parents and children by modelling the mediating/moderating roles of child number on the relations among parenting stress, parent-child literacy activities, and children’s reading interest.”, or “In this study, we operationalised implementing the Three-Child Policy as a macrosystem-level change that would impact the innermost microsystem”), yet this policy was introduced only in 2021, therefore the reform in itself cannot be studied. What authors actually study is how three children can potentially influence household life in light of policies that are brought forward. Therefore, It is important to distinguish between the two, as the actual three child policy may bring forward certain improvements that would make the life of parents more balanced, counteracting these conclusions. We can, however, see and evaluate this in ten years or so. What authors find now is a result of past policies or lack of them and would not be reliable for forecasting.
Second, the association between parenting quality and cognitive developments should be better discussed as a premise. The same applies to parenting stress mechanisms which should be better explained. Below are some useful references:
Kulic, N., Skopek, J., Triventi, M., & Blossfeld, H. P. (2019). Social background and children's cognitive skills: The role of early childhood education and care in a cross-national perspective. Annual Review of Sociology, 45, 557-579 (and its references)
Lareau A. 2003. Unequal Childhoods. Class, Race, and Family Life. Oakland: Univ. Calif. Press
Third, the theoretical background can be improved in the whole article. I invite authors to base their hypotheses on literature: why would you expect that the number of children matters for parenting stress? Why would child number moderate the stress? I suggest that research questions are kept separate from the hypotheses, and that hypotheses (H1-H6) are managed in a separate section where all of them are backed up by the appropriate literature.
Fourth, convenience sampling approach was used in the analysis. I would suggest that the authors provide more details on possible quotas in the sample and sampling bias. Would it be possible to weight the results to make them more generalizable to the whole city or country? On that matter, it would be useful to report how the proportions found in the sample differ from the real population proportions on major characteristics of parents (education, age, occupation).
Fifth, hypotheses that rely on regression analyses need to stipulate that the statements apply after accounting for controls (H3-6).
Also, I would like to see full regression tables of the results in the text.
Could the authors clarify whether both models 3 and 4 are based on specifications with the same control variables? I thought this was the case, but then the authors write only for H4 “In addition, we added SES, children’s age and gender as control variables to the model.” Does this mean that H3 does not include these controls?
Author Response
Response to Reviewer 1 Comments
I compliment the authors for an interesting article. The study examines the relationship between parenting stress, parent child literacy activities and the reading interest of children. It particularly focuses on the moderating role of the number of children in the Chinese context. Below I provide some suggestions that go in the direction of further improvement before the publication.
Our Responses: Thank you for your constructive suggestion! We have considered the comments carefully and made the corrections accordingly.
My first suggestion would be to move from the focus on three-child policy. I found it misleading to look at the article as an evaluation of this new measure of Chinese Government. In other words, the authors indicate that they study the effect of three child policy(“Therefore, this study 96 empirically examined the Three-Child Policy’s effect on Chinese parents and children by modelling the mediating/moderating roles of child number on the relations among parenting stress, parent-child literacy activities, and children’s reading interest.”) or “In this study, we operationalised implementing the Three-Child Policy as a macrosystem-level change that would impact the innermost microsystem”), yet this policy was introduced only in 2021, therefore the reform in itself cannot be studied. What authors actually study is how three children can potentially influence household life in light of policies that are brought forward. Therefore, it is important to distinguish between the two, as the actual three child policy may bring forward certain improvements that would make the life of parents more balanced, counteracting these conclusions. We can, however, see and evaluate this in ten years or so. What authors find now is a result of past policies or lack of them and would not be reliable for forecasting.
Our Responses: Thanks for your insightful comments and constructive suggestions. In the revision, we have moved from the focus on the three-child policy and examined how the increased number of children may potentially influence Chinese families and child development.
Second, the association between parenting quality and cognitive development should be better discussed as a premise. The same applies to parenting stress mechanisms which should be better explained. Below are some useful references:
Kulic, N., Skopek, J., Triventi, M., & Blossfeld, H. P. (2019). Social background and children's cognitive skills: The role of early childhood education and care in a cross-national perspective. Annual Review of Sociology, 45, 557-579 (and its references)
Lareau A. 2003. Unequal Childhoods. Class, Race, and Family Life. Oakland: Univ. Calif. Press
Our Responses: Thank you for your detailed suggestions! After carefully reading the references you provided and other relevant articles, we have further discussed the association between parenting quality and cognitive development and parenting stress mechanisms in this revision.
Third, the theoretical background can be improved in the whole article. I invite authors to base their hypotheses on literature: why would you expect that the number of children matters for parenting stress? Why would child number moderate the stress? I suggest that research questions are kept separate from the hypotheses, and that hypotheses (H1-H6) are managed in a separate section where all of them are backed up by the appropriate literature?
Our Responses: Thanks a lot for this constructive suggestion. In this revision, we have presented our hypotheses (H1-H6) in a separate section and developed our hypotheses from the appropriate literature.
Fourth, convenience sampling approach was used in the analysis. I would suggest that the authors provide more details on possible quotas in the sample and sampling bias. Would it be possible to weight the results to make them more generalizable to the whole city or country? On that matter, it would be useful to report how the proportions found in the sample differ from the real population proportions on major characteristics of parents (education, age, occupation).
Our Responses: Thanks for this constructive suggestion. We recruited the participants from the selected three kindergartens in the city, which represented a wide range of socioeconomic status. This is because they were selected from different levels of quality (“Municipal Demonstration Kindergarten” or “not rated”) and different areas (urban area or rural area). In this revision, we have provided more information about this. However, we have to admit that the generalizability of the results to the whole city or country might be limited; thus, we have addressed this limitation and future research directions in this revision.
Fifth, hypotheses that rely on regression analyses need to stipulate that the statements apply after accounting for controls (H3-6).
Also, I would like to see full regression tables of the results in the text.
Our Responses: Thanks a lot for your concern. In the “Results” section of this R1, we have reported the statements of regression analyses after accounting for controls. And we have provided full regression tables of the results in this revision. Thanks!
Could the authors clarify whether both models 3 and 4 are based on specifications with the same control variables? I thought this was the case, but then the authors write only for H4 “In addition, we added SES, children’s age and gender as control variables to the model.” Does this mean that H3 does not include these controls?
Our Responses: Thank you for your suggestion! Both models 3 and 4 are based on specifications with the same control variables (child age, child gender, parent gender, and SES). We added the note “In each model child age, child gender, parent gender, and SES were as control variables entered in first step” in tables 3 and 4 in this revision.

Reviewer 2 Report
Thank you for the opportunity to revise this paper. Despite having a large sample, I cannot understand the pertinence of the study. The variables under study are obvious, in my opinion. I have a hard time understanding what it adds to literature. I do not believe that the work is sufficiently mature to be published.
Major Points:
Abstract: Although the writing is clear, a background should have been introduced, The authors start with the objective abruptly.
Introduction
1. From the point of view of form, I think the ideas are little articulated.
2. The introduction is mostly taken by obvious, tested and scientifically proven statements. The relationships between variables also seem obvious and weakly supported by Bronfenbrenner's theory and the FST perspective. It does not seem to me that the mediational model meets these theories.
3. For example, I don't understand why the mediator is parent-child literacy activities - what model is this based on? because it is not parental stress that mediates the relationship between activities and interest in reading.
- The research relevance of the study is not addressed. I would like to have seen a greater problematization regarding the pertinence of this study. I do not see to what extent this study increases scientific knowledge to the Psychology.
Methods:
1. The description of the participants is poor. There is a lack of information, namely, data about neurological, psychological, and psychiatric disorders (specially of the parents), use of medication…
2. The methods section is also disorganized in places and lacking in detail. Paragraph structure seems haphazard in places (in general, one-sentence paragraphs should be avoided).
3. I can't figure out what age children start to read in China. Can 5-year-olds read?
4. The adaptation and statistical procedure seem to me to be adequate.
5. Concerning procedure, I believe that you followed the recommendations of Helsinki, but this are omitted.
6. Does this study not have the opinion of an ethics committee?
7. We know that in parental studies the mother's influence is not the same as that of the father. So here this has not been controlled, which is problematic. The effect of the parents' gender is decisive and could make a greater contribution. There is also a limitation, which is the fact that mothers respond on behalf of the fathers whenever the questionnaires are online.
Results
1. They are clear and appear conveniently described.
Discussion:
- I think the discussion is very weak. One of the reasons, I think, is that there is no theoretical background in the introduction that helps us discuss the results.
- The discussion is nothing more than a set of results that intend to answer the hypotheses without the least problematization with the literature. Why stress can influence the relationship between reading activities and reading interest has never been explored.
Author Response
Response to Reviewer 2 Comments
Thank you for the opportunity to revise this paper. Despite having a large sample, I cannot understand the pertinence of the study. The variables under study are obvious, in my opinion. I have a hard time understanding what it adds to literature. I do not believe that the work is sufficiently mature to be published.
Our Responses: We appreciate your critical and insightful comments! We hope our careful revisions meet your expectations.
Abstract
Although the writing is clear, a background should have been introduced, The authors start with the objective abruptly.
Our Responses: Thanks for your constructive suggestion. In this R1, we have added the research background of this study in the abstract.
Introduction
- From the point of view of form, I think the ideas are little articulated.
Our Responses: We rewrote the introduction to highlight the theories of this study.
- The introduction is mostly taken by obvious, tested and scientifically proven statements. The relationships between variables also seem obvious and weakly supported by Bronfenbrenner's theory and the FST perspective. It does not seem to me that the mediational model meets these theories.
Our Responses: We added some potions in the introduction to strengthen the theoretical base of this research model and better justify this study.
- For example, I don't understand why the mediator is parent-child literacy activities - what model is this based on? because it is not parental stress that mediates the relationship between activities and interest in reading.
Our Responses: Thank you for this critique. Our rationale is: FST emphasizes family functioning as a major force contributing to child development. FST and ecological systems theory view different levels of social-ecological influences on children’s development through the activities in which children engage. Children can learn and develop only through activities with others or children’s interaction patterns. From this perspective, we examined parent-child literacy activities occurring within the family as the mediating factor in the relationship between parenting stress and children’s literacy development.
The research relevance of the study is not addressed. I would like to have seen a greater problematization regarding the pertinence of this study. I do not see to what extent this study increases scientific knowledge to the Psychology.
Our Responses: Thanks for your critical comments. In this R1, we have addressed your concerns about the unique theoretical and practical implications of this work, especially in the last two paragraphs of the section “Limitations, Future Directions, and Implications”.
Method
- The description of the participants is poor. There is a lack of information, namely, data about neurological, psychological, and psychiatric disorders (specially of the parents), use of medication…
Our Responses: Thanks for your insightful comments. Sample characteristics based on parental responses to Parenting Stress Index were largely consistent with characteristics expected in a community-based sample rather than a clinic sample. In this study, the children and their parents were not experiencing any major psychological or psychiatric disorders. Nevertheless, we have added this information in R1.
- The methods section is also disorganized in places and lacking in detail. Paragraph structure seems haphazard in places (in general, one-sentence paragraphs should be avoided).
Our Responses: Thanks for your constructive suggestion. We have refined the methods section to improve its quality and readability, cleaning all the one-sentence paragraphs.
- I can’t figure out what age children start to read in China. Can 5-year-olds read?
Our Responses: Thanks for your inspiring question. Yes, in Mainland China, children under Age 6 are not allowed to read and write formally, which is ‘formal literacy’ and should be banned in preschools and kindergartens (Li, 2014). But young children are encouraged to ‘read’ picture books that have no or few Chinese characters with their parents and teachers, which is called ‘informal literacy’ (Li, 2014) . Thanks.
Reference: Li, H.(2014). Teaching Chinese Literacy in the Early Years: Psychology, Pedagogy, and Practice. U.K.: Routledge.
- The adaptation and statistical procedure seem to me to be adequate.
Our Responses: Thank you!
- Concerning procedure, I believe that you followed the recommendations of Helsinki, but this are omitted.
Our Responses: We have added the statement about the ethics issue in the revision.
- Does this study not have the opinion of an ethics committee?
Our Responses: Thank you for your careful review! We clarified the ethics issue in the last paragraph in the sub-section of 2.1 Participants and Procedure.
- We know that in parental studies the mother’s influence is not the same as that of the father. So here this has not been controlled, which is problematic. The effect of the parents’gender is decisive and could make a greater contribution. There is also a limitation, which is the fact that mothers respond on behalf of the fathers whenever the questionnaires are online?
Our Responses: Thank you for your suggestion! In this revision, we have added parent gender as the control variable in the model.
Results
- They are clear and appear conveniently described.
Our Responses: Thank you for this favorable comment!
Discussion
- I think the discussion is very weak. One of the reasons, I think, is that there is no theoretical background in the introduction that helps us discuss the results.
Our Responses: Thank you for this comment! In this revision, we added theoretical background in the introduction to strengthen the significance of this work.
- The discussion is nothing more than a set of results that intend to answer the hypotheses without the least problematization with the literature. Why stress can influence the relationship between reading activities and reading interest has never been explored.
Our Responses: Thank you so much for this insightful comment! Note that we didn’t examine parenting stress as a mediator in the relationship between reading activities and reading interest. Rather, informed by the ecological systems theory and Family Systems Theory (FST), we examined the moderating role of increased child number and the mediation role of parent-child literacy activities in the relationship between parenting stress and young children’s reading interest. There are specific portions of this manuscript that address the reviewer’s concern about the unique theoretical and practical implications of this work in the last two paragraphs in the section on Limitations, Future Directions, and Implications.

Reviewer 3 Report
The paper presents an important issue, especially in the Chinese context. The presentation of the reference literature is complete. The methods, the research design and the data analysis are presented clearly, and the authors give enough information about them. The presentation of results is satisfactory, the statistical analysis is correct, and the applied statistical analysis is appropriate. The discussion is fleshed out and related to the literature review and emphasises the contribution of this important study.
In the discussion paragraph, the authors write: "we suggest that [...] parents should attempt to reduce their parenting stress" - but they don't write, how. Who and how can help parents to reduce stress? Which sector or what kind of professional? I think the authors should give some suggestions for these questions.
In general, the work makes a positive impression. I wish the authors all the best in progressing their work.
Author Response
Response to Reviewer 3 Comments
The paper presents an important issue, especially in the Chinese context. The presentation of the reference literature is complete. The methods, the research design and the data analysis are presented clearly, and the authors give enough information about them. The presentation of results is satisfactory, the statistical analysis is correct, and the applied statistical analysis is appropriate. The discussion is fleshed out and related to the literature review and emphasises the contribution of this important study.
Our Responses: Thanks a lot for your favorable comments!
In the discussion paragraph, the authors write: “we suggest that [...] parents should attempt to reduce their parenting stress” - but they don't write, how. Who and how can help parents to reduce stress? Which sector or what kind of professional? I think the authors should give some suggestions for these questions.
Our Responses: Thanks a lot for your insightful comments and constructive suggestions. In this R1, we have elaborated more on the practical implications of this study in the section of Limitations, Future Directions, and Implications.
In general, the work makes a positive impression. I wish the authors all the best in progressing their work.
Our Responses: Thank you again for your encouragement!

Round 2
Reviewer 1 Report
Thank you for your revisions, which have sufficiently responded to my concerns. I believe that the manuscript has been improved. I only have a minor request. Could you please report the official statistics for China regarding the percentage of families with 1, 2 or 3 children for the year of your analyses? If possible, it would be great to have this distinction (1,2, 3 children) by education of parents too. You could report on this either in the data section or in the introduction.
Author Response
Response to Reviewer 1 Comments
Thank you for your revisions, which have sufficiently responded to my concerns. I believe that the manuscript has been improved. I only have a minor request. Could you please report the official statistics for China regarding the percentage of families with 1, 2 or 3 children for the year of your analyses? If possible, it would be great to have this distinction (1,2, 3 children) by education of parents too. You could report on this either in the data section or in the introduction.
Our Responses: Thank you! We have reported in the introduction about the latest official statistics for China regarding the percentage of families with 1, 2 or 3 children. Since the official statistics have reported only maternal education in families with different numbers of children, we have reported maternal education levels with this distinction (1, 2, 3 children). In addition, we controlled the SES (including parental education) in the process of data analysis.

Reviewer 2 Report
Dear authors
Despite the improvement effort I still find the relevance of the study quite low.
Author Response
Response to Reviewer 2 Comment
Despite the improvement effort I still find the relevance of the study quite low.
Our Responses: We appreciate your comment. According to your suggestions in R1, we have strengthened the theoretical base and elaborated the associations among all the research variables, discussed the results of our research model based on relevant theoretical and existing studies, and discussed the limitations for the future research. We hope our careful revisions meet your expectations.
